# Risk of 30-day hospital readmission associated with medical conditions and drug regimens of polymedicated, older inpatients discharged home: a registry-based cohort study

Filipa Pereira [ID],[1,2] Henk Verloo [ID],[2,3] Zhivko Taushanov,[4] Saviana Di Giovanni,[2,5] Carla Meyer-Massetti,[6] Armin von Gunten,[3] Maria Manuela Martins,[1,7] Boris Wernli[8]

For numbered affiliations see end of article.

**Correspondence to**
Filipa Pereira;
filipa.pereira@hevs.ch

## ABSTRACT

**Objectives** The present study analysed 4 years of a hospital register (2015–2018) to determine the risk of 30-day hospital readmission associated with the medical conditions and drug regimens of polymedicated, older inpatients discharged home.

**Design** Registry-based cohort study.

**Setting** Valais Hospital—a public general hospital centre in the French-speaking part of Switzerland.

**Participants** We explored the electronic records of 20 422 inpatient stays by polymedicated, home-dwelling older adults held in the hospital's patient register. We identified 13 802 hospital stays by older adults who returned home involving 8878 separate patients over 64 years old.

**Outcome measures** Sociodemographic characteristics, medical conditions and drug regimen data associated with risk of readmission within 30 days of discharge.

**Results** The overall 30-day hospital readmission rate was 7.8%. Adjusted multivariate analyses revealed increased risk of hospital readmission for patients with longer hospital length of stay (OR=1.014 per additional day; 95% CI 1.006 to 1.021), impaired mobility (OR=1.218; 95% CI 1.039 to 1.427), multimorbidity (OR=1.419 per additional International Classification of Diseases, 10th Revision condition; 95% CI 1.282 to 1.572), tumorous disease (OR=2.538; 95% CI 2.089 to 3.082), polypharmacy (OR=1.043 per additional drug prescribed; 95% CI 1.028 to 1.058), and certain specific drugs, including antiemetics and antinauseants (OR=3.216 per additional drug unit taken; 95% CI 1.842 to 5.617), antihypertensives (OR=1.771; 95% CI 1.287 to 2.438), drugs for functional gastrointestinal disorders (OR=1.424; 95% CI 1.166 to 1.739), systemic hormonal preparations (OR=1.207; 95% CI 1.052 to 1.385) and vitamins (OR=1.201; 95% CI 1.049 to 1.374), as well as concurrent use of beta-blocking agents and drugs for acid-related disorders (OR=1.367; 95% CI 1.046 to 1.788).

**Conclusions** Thirty-day hospital readmission risk was associated with longer hospital length of stay, health disorders, polypharmacy and drug regimens. The drug regimen patterns increasing the risk of hospital readmission were very heterogeneous. Further research is

## STRENGTHS AND LIMITATIONS OF THIS STUDY

⇒ The records of 20 422 hospitalisations involving 8878 different polymedicated, home-dwelling, older patients readmitted to hospital at least once were studied to determine the risk of 30-day hospital readmission.

⇒ The study included 4-year data from a comprehensive hospital register (2015–2018).

⇒ A whole series of sociodemographic and clinical parameters, medical conditions, and prescribed drugs were used to predict the probability of hospital readmission.

⇒ Analyses were correlational and causality was not explored.

⇒ Although the study considered statistical associations between drugs and hospital readmissions, it did not consider clinically diagnosed drug–drug interactions.

needed to explore hospital readmissions caused solely by specific drugs and drug–drug interactions.

## INTRODUCTION

Longitudinal studies have demonstrated that approximately 20% of home-dwelling older adults supported by home healthcare services experienced hospital readmission within 30 days of their discharge.[1–3] For many older adults, readmission to an acute hospital is associated with a functional decline that has not always recovered by the time they are discharged.[4] However, the systematic review by Hansen *et al*[5] revealed wide-ranging estimates (5%–79%) of how many hospital readmissions were preventable. The period between hospital discharge and readmission has not always been clearly stated in the literature, ranging from 30 days to 3 years. However, 30 days is the most frequently used

in public health policy when measuring healthcare system performance.[6–8]

Numerous determinants have been identified and associated with hospital readmissions, for example, sociodemographic and individual characteristics, multimorbidity, and medical events.[9 10] A substantial risk of 30-day hospital readmission has been associated with older inpatients treated for different diseases and surgical interventions involving hip fracture, cancer, bypass, acute cardiovascular events or complex surgery.[11] The reasons for hospital readmission after a surgical intervention are often not directly related to the surgery itself but rather to underlying chronic health conditions.[12] Thus, chronic diseases may play an important role in readmission risk, independent of the reason for the initial hospitalisation.[13 14] Older adults' chronic diseases are not isolated health conditions; they can influence each other, and treatment for one disease may adversely affect another.[15] For all these reasons, patterns of 30-day hospital readmissions may be very complex.[16]

Multimorbidity, in the case of two or more diseases,[17 18] may require taking multiple medicines,[19] known as polypharmacy (PP) when daily intake involves five or more drugs.[20] Increasing incidences of multimorbidity with age, and consequently PP, add to the complexity of managing older inpatients' drug prescriptions, particularly at hospital discharge.[21 22] PP and inadequate drug management are significant risk factors for adverse drug events (ADEs)—the most common postdischarge complications—alongside hospital-acquired infections and procedural complications.[23 24] ADEs resulting from inappropriate drug prescribing, discrepancies between prescribed and current regimens, poor adherence, and the inadequate surveillance of adverse effects frequently lead to hospital admissions, readmissions[8] and other undesirable consequences such as increased morbidity, decreased autonomy, institutionalisation and even early death.[25 26] A systematic review by El Morabet et al[27] indicated ADE rates of 18%–38% after hospital discharge and 4.5%–24% hospital readmission rates due to those ADEs. Because older adults use more drugs, they are at a greater risk of drug-related readmission. Numerous studies have found that nearly 30% of older inpatients experienced ADEs within 3 weeks of hospital discharge, almost three-quarters of which could have been prevented or lessened.[10 28 29]

Despite the significant overall impact of ADEs on hospital readmission rates, little is known about the association of hospital readmission risk with medical conditions and drug regimens.[30 31] El Morabet et al revealed the high prevalence of antibiotics, diuretics, vitamin K antagonists, opioids, antidiabetics, anticancer drugs, antihypertensives, digitalis glycosides, corticosteroids and psychotropic drugs in drug-related hospital readmissions.[27] Samoy et al[32] reported that anticoagulants, hypoglycaemics, beta-blocking agents, antidepressants, calcium channel blockers and lenograstim were associated with high risk of hospital readmission. A retrospective patient record study by Teymoorian et

al[33] reported that anticoagulants and antiplatelet agents, diuretics and antihypertensives, and opioids were associated with a high risk of persons aged 80 years old or more being readmitted to hospital within 30 days. Blanc et al reported the readmission scores of different drugs in a large sample of 10 374 adult hospital admissions in general medicine. Taking beta-blocking agents, calcium channel blockers, diuretics, hypoglycaemic drugs or opioids was a significant risk for 30-day readmission.[9]

Besides the higher risk of drug-related hospital readmission, some studies have also investigated the associations between combining drugs—a common practice when treating complex diseases or coexisting medical conditions—and drug-related hospital readmissions. Although using multiple drugs may be good clinical practice and compliant with guidelines for treating certain diseases, one significant consequence of combining drugs is that patients face much higher risk of ADEs, which can be caused by drug–drug interactions.[34–36] ADEs can emerge because a drug's pharmacokinetics and pharmacodynamics change if taken with another drug.[36] Moura et al[37] found that participants with potential drug–drug interactions on their prescription list had 2.4 times higher adjusted OR of being readmitted.

Even though some studies have reported high numbers of readmissions among home-dwelling older patients for a variety of drugs,[38] this health issue was mostly investigated using prospective or cross-sectional studies with small samples. More insight is needed into the patterns of drug-related hospital readmissions and risk factors in order to design better interventions for addressing ADEs.[39 40] As part of a broader project,[41] the present study's goal was to use hospital register data to prioritise risk factors for hospital readmission. We hypothesised that sociodemographic characteristics, medical conditions (defined using the WHO's International Classification of Diseases, 10th Revision (ICD-10) and the Swiss Classification of Surgical Interventions (CHOP)) and drug prescriptions (based on the WHO's Anatomical Therapeutic Chemical (ATC) Classification System) were significant risk factors for 30-day hospital readmission for discharged older adults.

## MATERIALS AND METHODS
### Study design
This longitudinal study (2015–2018) used data on a population cohort taken from a hospital registry composed of 140 variables. These were used to investigate the associations between risk of 30-day hospital readmission and the medical conditions and drug regimens of polymedicated older inpatients discharged home. The study was performed with close regard to the RECORD (REporting of studies Conducted using Observational Routinely collected health Data) statement.[42]

### Population and data collection
Our custom, 4-year, registry-based data set included polymedicated inpatients (five or more drugs prescribed at

hospital discharge), aged 65 years old or more, living in their own homes and hospitalised at least once at the Valais Hospital (a public general hospital in the French-speaking part of Switzerland). This specific population was selected due to its increased risk of hospital readmission.[10 28 29] Older inpatients hospitalised once only or who died during hospitalisation were excluded, as were those hospitalised for fewer than 24 hours (the criterion to count as 'hospitalised' in Switzerland). Valais Hospital's register contains a comprehensive electronic health record composed of 140 variables routinely collected during hospital stays. However, no electronic patient records were available for adult psychiatry for 2015–2018. The extracted patient data contained sociodemographic characteristics, medical and surgical diagnoses, and routinely assessed clinical data (such as gait, falls risk or hearing) from hospitalised patients with at least five prescribed drugs at discharge. Medical and surgical diagnoses were coded based on the ICD-10 and CHOP.[43] Drug classification was based on the WHO's ATC Classification System.[44]

The strategy for transforming and synthesising the data extracted from the register's multiple data set sources was based on Olsen's register-based methodological considerations[45] and has been documented elsewhere.[46] Our data set was composed of 20 422 hospital admission records running from January 2015 to December 2018, with similar numbers of annual hospital admissions: 5134, 5095, 5125 and 5068, respectively.

### Data set customisation for predictive analysis

The data set was recoded and customised to identify the frequency of older patients' hospital admissions. Each subject's unique identifier was used to distinguish their different hospital stays from 2015 to 2018. The data set included 13 802 hospital stays involving 8878 different older inpatients discharged home, and whose data were complete (no missing values).

Sociodemographic and clinical data were considered independent variables and used to compute the predictive models. Readmission following discharge home was defined as the dependent variable of interest and was dichotomised (0=no, 1=yes) based on 30-day readmission between 2015 and 2018. Furthermore, the custom data set was composed of six clinical clusters based on agglomerative hierarchical clustering methods for identifying clinically relevant characteristics and representing older inpatients' health status. Medical status and drugs data were recoded and copied to an exploitable population database.[46]

### Sociodemographic variables and length of stay

The sociodemographic data set—almost exclusively composed of ordinal variables—included two categorical variables (sex and place of discharge from hospital) and three continuous variables (age and admission and discharge dates). Sex and age were included in the analysis as sociodemographic control variables. Age was considered

a continuous variable as its progressive impact has been proven in preliminary investigations and previous studies.[47]

### Health variables

Numerous variables were used to describe older patients' health status during each hospital stay. The health data set was composed of 23 categorical variables: 21 measured as ordinal variables (mobility, changing position, falls in the last year, etc) and two measured as nominal variables (altered gait and chronic pain). A cleaner, better-structured data set—composed of hierarchical clusters—was obtained in a previous study combining empirical and best-practice statistical approaches.[46] Three of six preliminarily computed hierarchical clusters were included in the modelling analysis as confounding variables: the mobility cluster, the dependency on the activities of daily living cluster and the mental state cluster.[46] These three clusters were selected because of their significant contributions to hospital readmissions.[48–50] The data set of medical information was composed of patients' principal medical diagnosis and four secondary medical diagnoses, based on ICD-10. Finally, the year of hospitalisation was introduced as a control variable, based on the fact that earlier admission to hospital during this period led to a higher probability of unplanned readmissions during the entire period covered.

### Included drugs

The hospital data set showed that discharged patients had been prescribed 2370 different drugs. Drug prescriptions were considered continuous, classified according to the WHO's ATC Classification System[51] and then included in the predictive model as independent variables. To ensure robust statistical results, the model only included drug categories prescribed to at least 30 inpatients who were readmitted within 30 days. Online supplemental file 1 presents the prescribed ATC classified drugs included in the predictive model as independent variables.

For statistical purposes, drug–drug interactions between different ATC drug classes[51] were operationalised as dichotomised variables (0=no simultaneous use of drugs from both classes, 1=simultaneous use of drugs from both classes) and added to the previous model. Drug class interactions were selected based on a literature review, significant ORs and expert opinions.[52]

### Data analysis strategy

Data were extracted into a Microsoft Excel spreadsheet (Microsoft, Redmond, Washington, USA) and then imported into SPSS V.26.0 software. We examined statistical associations between hospital readmissions and patient age and sex, length of stay (LOS), principal and related ICD-10 diagnoses, CHOP interventions, and drug prescriptions during hospitalisations. A causality analysis between those variables was impossible given our retrospective data collection method, our inability to calculate the time between drug intake and readmission, and the potential drug changes between hospitalisation

sequences. We conducted a bivariate analysis relating the independent variables to 30-day readmission after discharge home from 2015 to 2018. Next, we calculated a series of multilevel logistic regression models for binary outcomes explaining the readmissions, within 30 days, of patients discharged home (0=no, 1=yes). These hierarchical models included two levels: the first level concerned hospital stays themselves, nested in the second level, that of individuals. First, we computed a baseline multilevel binary logistic regression model to estimate how sets of predictors influenced the probability of 30-day hospital readmission, which included individuals' characteristics, health conditions and hospital LOS. Second, we completed this baseline model with the drugs prescribed to older inpatients on their discharge home. Finally, to the baseline model completed with prescribed drugs, we added the known drug–drug interactions between different ATC drug classes, based on a literature review and expert opinions. The model computed each predictor's impact, other things being equal, by estimating its net impact, controlling for other factors (adjusted ORs). The model also considered correlations between each subject's different variables, which were generally not independent.[53] The model's random intercept design allowed each individual's intercept to vary, assuming that some unmeasured traits remained stable over time and allowing a better estimation of the model's parameters. The estimated parameters, on the other hand, had the same effect on every subject. Since the data were based on the whole population—not a sample—of polymedicated older inpatients discharged home from the Valais Hospital, the ORs' CIs and statistical tests were used to indicate the robustness of the relationships (they usually only make sense for statistical inference).

### Patient and public involvement

Patients were not involved in the development of the research questions, study design, outcome measures and conduct of the study.

### RESULTS
### Descriptive results

The electronic records of 20 422 inpatient stays by polymedicated, home-dwelling older adults included a total of 13 802 hospital stays of 8878 different older inpatients who returned home and whose data were all non-missing–an average of 1.55 inpatient hospital readmissions. The total sample's mean age was 77.77 years old (SD=7.48) and 57% were men (table 1). The average hospital LOS was 8.44 days (SD=7.58). At discharge, 25% of the sample had impaired mobility, 4% were impaired in their activities of daily living and 4% showed mental impairment. Our sample population averaged 4.58 (SD=0.92) ICD-10 diagnoses and 1.83 (SD=1.76) surgical interventions (CHOP) performed during hospitalisation. The selected medical diagnoses distinguished patients affected by circulatory (24%), infectious (3%) and respiratory (11%) diseases, as well as trauma (8%) and tumours

**Table 1** Sociodemographic and hospitalisation data for inpatient stays by polymedicated, home-dwelling adults aged 65 or more (N=13 802)

| Variables | Inpatient stays by polymedicated, home-dwelling adults aged 65 or more (N=13 802) |
|---|---|
| Sex | |
| Stays by men (%) | 7834 (56.8) |
| Stays by women (%) | 5968 (43.2) |
| Age at discharge (years) | |
| Mean inpatient age at discharge (SD) | 77.77 (7.48) |
| Minimum–maximum | 65–106 |
| Median (IQR 25–75) | 77.00 (68.00–80.00) |
| 65–69 (%) | 2226 (16.1) |
| 70–79 (%) | 5811 (42.1) |
| 80–89 (%) | 4845 (35.1) |
| 90 and more (%) | 920 (6.7) |
| Year of discharge | |
| 2015 (%) | 3501 (25.4) |
| 2016 (%) | 3318 (24.0) |
| 2017 (%) | 3530 (25.6) |
| 2018 (%) | 3453 (25.0) |
| Length of stay (days) | |
| Mean (SD) | 8.44 (7.58) |
| Minimum–maximum | 1–149 |
| Median (IQR 25–75) | 7 (4–11) |
| Number of ICD-10 conditions | |
| Mean (SD) | 4.58 (0.92) |
| Minimum–maximum | 1–5 |
| Median (IQR 25–75) | 5 (5–5) |
| Principal ICD-10 diagnosis | |
| Circulatory (%) | 3336 (24.2) |
| Infectious (%) | 404 (2.9) |
| Respiratory (%) | 1444 (10.5) |
| Trauma (%) | 1043 (7.6) |
| Tumours (%) | 1505 (10.9) |
| Number of CHOP surgical procedures | |
| Mean (SD) | 1.83 (1.76) |
| Minimum–maximum | 0–5 |
| Median (IQR 25–75) | 1 (0–3) |
| Number of medicines prescribed at hospital discharge | |
| Mean (SD) | 8.95 (3.24) |
| Minimum–maximum | 5–30 |
| Median (IQR 25–75) | 8 (7.50–16.00) |

CHOP, Swiss Classification of Surgical Interventions; ICD-10, International Classification of Diseases, 10th Revision.

(11%). On average, 8.95 (SD=3.24) drugs were prescribed per patient at hospital discharge.

### Associations between 30-day hospital readmission risk and sociodemographic characteristics and medical conditions

The rate of 30-day hospital readmission for older patients discharged home was 7.8%. Bivariate associations with $\chi^2$

tests showed significant differences between older inpatients' sociodemographic characteristics and medical conditions (table 2). Men showed a slightly higher proportion of 30-day hospital readmissions than women (8.2% vs 7.3%). However, age did not significantly affect the probability of 30-day readmission. More readmissions were also seen among older patients with a circulatory disease (8.2% vs 6.5%), those not affected by trauma (8.0% vs 5.8%) and especially those with a tumour (15.1% vs 6.9%). Multimorbidity also increased the risk of 30-day hospital readmissions—from 1.5% for older patients with a single ICD-10 condition to 8.8% for those with five.

## Associations between 30-day hospital readmission risk and drugs

On average, older patients readmitted within 30 days had more prescribed drugs than those who were not readmitted (9.95 vs 8.87 drugs). We found a linear relationship between the 30-day readmission rate and the average number of prescribed drugs (p>0.001), which supported the absence of a cut-off point in this relationship (figure 1).

Among the most robust statistical associations ($\chi^2$ tests) with 30-day hospital readmissions involved the classes of drugs including antineoplastics and immunomodulators (12.6% vs 7.6% for those not treated with them) and antiemetics and antinauseants (27.7% vs 7.7%). There was also a higher risk of 30-day hospital readmission among older inpatients taking drugs for functional gastrointestinal disorders (13.4% vs 7.4%) and antihypertensives (14.1% vs 7.7%) (table 3).

## Baseline multivariate model

A baseline multivariate logistic regression model including older patients' sociodemographic and clinical variables, but not their prescribed drugs at discharge, was computed to predict 30-day hospital readmission after discharge home (table 4). Neither sex nor age had a significant impact. On the contrary, LOS had a significant impact (OR=1.014 for each additional day; 95% CI 1.006 to 1.021), as did mobility (OR=1.218 for older patients with an impaired mobility status; 95% CI 1.039 to 1.427). Dependence on the activities of daily living and mental health status showed no influence. Concerning diagnoses measured in the ICD-10, we found that older patients with a tumorous disease (OR=2.538; 95% CI 2.089 to 3.082) were much more susceptible to 30-day hospital readmission. Patients with circulatory pathologies showed no difference from the reference category (OR=0.938; 95% CI 0.783 to 1.124), nor did those with respiratory problems (OR=1.100; 95% CI 0.875 to 1.382), trauma (OR=0.847; 95% CI 0.633 to 1.134) or infection-related problems (OR=1.381; 95% CI 0.964 to 1.977; p=0.078). Multimorbidity predicted a higher probability of readmission (OR=1.419 per additional ICD-10 condition; 95% CI 1.282 to 1.572), whereas the number of surgical procedures had no noticeable impact (OR=0.978; 95% CI 0.938 to 1.020). The year of hospital stay did have an impact,

**Table 2** 30-day hospital readmission risk at different periods for different age groups (N=13 802)

| Variables | 30-day hospital readmission (%) | P value |
|---|---|---|
| Complete sample | 7.8 | |
| Sex | | * |
| Female vs male | 7.3 vs 8.2 | |
| Year-end age, in years | | NS |
| 65–69 | 7.5 | |
| 70–79 | 7.6 | |
| 80–89 | 8.4 | |
| ≥90 | 6.4 | |
| Mobility cluster | | NS |
| Preserved mobility vs impaired mobility | 7.6 vs 8.5 | |
| Activities in daily living (ADL) | | NS |
| Full ADL ability vs impaired ADL | 7.8 vs 7.2 | |
| Cognitive status | | NS |
| Preserved cognitive status vs cognitive impairment | 7.8 vs 7.9 | |
| ICD-10 diagnosis: circulatory problems | | ** |
| No vs yes | 8.2 vs 6.5 | |
| ICD-10 diagnosis: infection | | NS |
| No vs yes | 7.7 vs 9.9 | |
| ICD-10 diagnosis: respiratory problems | | NS |
| No vs yes | 7.8 vs 8.0 | |
| ICD-10 diagnosis: trauma | | ** |
| No vs yes | 8.0 vs 5.8 | |
| ICD-10 diagnosis: tumour | | *** |
| No vs yes | 6.9 vs 15.1 | |
| Number of ICD-10 conditions | | *** |
| 1 | 1.5 | |
| 2 | 4.9 | |
| 3 | 3.6 | |
| 4 | 4.8 | |
| 5 | 8.8 | |
| Number of surgical procedures (CHOP) | | * |
| 0 | 7.7 | |
| 1 | 7.8 | |
| 2 | 7.0 | |
| 3 | 7.3 | |
| 4 | 7.1 | |
| 5 | 9.7 | |
| Year of discharge: 2015–2018 | | NS |
| 2015 | 8.3 | |
| 2016 | 8.0 | |
| 2017 | 8.0 | |
| 2018 | 6.8 | |

*P<0.05, **P<0.01, ***P< 0.001.
CHOP, Swiss Classification of Surgical Interventions; ICD-10, International Classification of Diseases, 10th Revision; NS, non-significant.

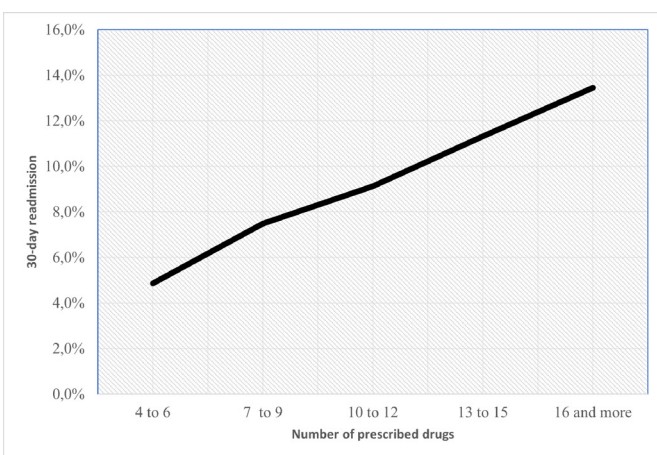

**Figure 1** Relationship between 30-day readmission rate and the number of prescribed drugs at discharge.

however, as the earlier the hospitalisation during the 4 years under review, the higher the probability of readmission (OR=0.933 per additional year; 95% CI 0.880 to 0.990).

Some variables that were non-significant in bivariate analyses became significant in multivariate analyses. This was because the results of multivariate analyses were controlled by all the other parameters and interpretations were made with 'other things being equal'. Also, the composition of subgroups could be very different in some bivariate analyses.

### Prediction of 30-day hospital readmission and drug prescriptions

Table 5 presents the baseline logistic regression model completed with the drugs prescribed to older patients at discharge home that were significantly associated (p≤0.05) with 30-day hospital readmission. It was not possible to introduce the total number of drugs prescribed jointly in this model because of their collinearity with other drug variables. Non-significant drugs and other variables have been omitted from table 3 in order to simplify the presentation. The probabilities of 30-day hospital readmission are presented in descending order of discharged older patients' ORs for each additional unit of the drugs in question. Intake of antiemetics and antinauseants was very strongly linked to 30-day readmission (OR=3.216 for each additional drug unit taken; 95% CI 1.842 to 5.617), as were those of antihypertensives (OR=1.771; 95% CI 1.287 to 2.438), gastrointestinal drugs (OR=1.424; 95% CI 1.166 to 1.739), systemic hormonal preparations (OR=1.207; 95% CI 1.052 to 1.385) and vitamins (OR=1.201; 95% CI 1.049 to 1.374). On the contrary, the intake of lipid-modifying agents was associated with a decrease in 30-day hospital readmissions (OR=0.841 for each drug from this class prescribed; 95% CI 0.732 to 0.967).

### Drug interactions and 30-day hospital readmissions

The model considered drug class interactions for the (1) cardiovascular system*central nervous system, gastrointestinal system, and metabolism*cardiovascular system; (2) gastrointestinal system and metabolism*central nervous system; (3) cardiovascular system*anti-infectives; and (4) central nervous system*anti-infectives. The analysis was carried out controlling for the basic model's variables (table 4), and the table reports the ORs for each additional unit of the statistically significant drugs in question as well as for significant drug interactions. Antiemetics and antinauseants were very strongly associated with 30-day readmission (OR=3.222; 95% CI 1.844 to 5.630), as were drugs regulating the gastrointestinal tract (OR=1.428; 95% CI 1.169 to 1.744) and systemic hormones (OR=1.210; 95% CI 1.054 to 1.390). The joint intake of beta-blocking agents and drugs for acid-related disorders was significantly associated with 30-day hospital readmission (OR=1.367; 95% CI 1.046 to 1.788); this is the only significant drug interaction in table 4. On the contrary, lipid-modifying agents were associated with lower 30-day hospital readmission (OR=0.838), as were substances acting on the renin–angiotensin system (OR=0.892; 95% CI 0.796 to 0.999) (table 6).

## DISCUSSION

The present study examined the records of 20 422 hospitalisations involving polymedicated home-dwelling older patients, eventually discharged home, to identify the risk of 30-day hospital readmission. These records were obtained from 4-year data of a comprehensive hospital register. The 8878 individual older patients readmitted to the Valais Hospital showed a 30-day hospital readmission rate of almost 8%, corroborating previously published all-cause hospital readmission rates among home-dwelling older patients.[9 27] However, Jencks et al[3] found a much higher 30-day readmission rate, reaching almost 20% among discharged older patients who had been hospitalised in acute medicine and surgery wards.[3] As a bivariate association, multimorbid men were at a significantly higher risk of readmission than multimorbid women; however, in the adjusted multivariate analysis, this significance disappeared. Medical conditions, PP and multiple classes of prescribed drugs were all associated with higher 30-day readmission rates, in line with previous studies.[27 54–56]

Our study found no significant differences in the risk of 30-day hospital readmission for men and women. However, some previous research found that men were more likely to forget to take their drugs or to not apply the changed drug dosages prescribed by their family physician, consequently increasing their risk of hospital readmission for drug-related problems.[57] Opposite results were found in a population-based study by Manteuffel et al,[58] with women being less likely than men to properly adhere to their drug prescriptions. These differences may indicate a need for more personalised drug prescription and drug management to improve clinical outcomes. Further research should explore associations between different types of drugs and sex,[58 59] but this topic was beyond the scope of the present study. Another interesting

**Table 3**  30-day hospital readmissions for different classes of drugs based on ATC (N=13 802)

| Drug class | 30-day readmission with no drugs in this class (%) | 30-day readmission with drugs in this class (%) | P value |
|---|---|---|---|
| First level, anatomical main group | | | |
| Blood and blood-forming organ drugs (B) | 7.1 | 8.0 | NS |
| Dermatologicals (D) | 7.7 | 9.4 | NS |
| Genitourinary system and sex hormones (G) | 7.7 | 8.3 | NS |
| Systemic hormonal preparations, excluding sex hormones and insulins (H) | 7.4 | 9.5 | *** |
| Anti-infectives for systemic use (J) | 8.0 | 7.2 | NS |
| Antineoplastic and immunomodulating agents (L) | 7.6 | 12.6 | *** |
| Drugs for the musculoskeletal system (M) | 8.0 | 6.5 | * |
| Antiparasitic products, insecticides and repellents (P) | 7.8 | 6.6 | *** |
| Respiratory system drugs (R) | 7.4 | 9.9 | *** |
| Sensory organ drugs (S) | 7.8 | 8.4 | NS |
| Second level, therapeutic subgroup | | | |
| Stomatological preparations (A01) | 7.8 | 12.2 | NS |
| Drugs for acid-related disorders (A02) | 7.0 | 8.5 | *** |
| Drugs for functional gastrointestinal disorders (A03) | 7.4 | 13.4 | *** |
| Antiemetics and antinauseants (A04) | 7.7 | 27.7 | *** |
| Bile and liver therapy drugs (A05) | 7.8 | 14.3 | NS |
| Drugs for constipation (A06) | 7.3 | 10.8 | *** |
| Antidiarrhoeals, intestinal anti-inflammatory/anti-infective agents (A07) | 7.7 | 12.9 | *** |
| Digestives, including enzymes (A09) | 7.8 | 10.0 | NS |
| Drugs used in diabetes (A10) | 7.4 | 9.5 | *** |
| Vitamins (A11) | 7.5 | 9.9 | *** |
| Mineral supplements (A12) | 7.4 | 8.8 | ** |
| Other alimentary tract and metabolism products (A16) | 7.8 | 6.3 | NS |
| Cardiac therapy (C01) | 7.6 | 8.9 | NS |
| Antihypertensives (C02) | 7.7 | 14.1 | *** |
| Diuretics (C03) | 7.2 | 9.8 | *** |
| Peripheral vasodilators (C04) | 7.8 | 15.2 | NS |
| Vasoprotective drugs (C05) | 7.8 | 9.8 | NS |
| Beta-blocking agents (C07) | 7.1 | 8.6 | *** |
| Calcium channel blockers (C08) | 7.7 | 8.6 | NS |
| Agents acting on the renin–angiotensin system (C09) | 8.7 | 7.1 | *** |
| Lipid-modifying agents (C10) | 8.3 | 7.1 | ** |
| Anaesthetics (N01) | 7.8 | 18.8 | * |
| Analgesics (N02) | 7.8 | 7.8 | NS |
| Antiepileptics (N03) | 7.7 | 9.0 | NS |
| Drugs for Parkinson's disease (N04) | 7.8 | 6.9 | NS |
| Psycholeptics (N05) | 6.8 | 9.3 | *** |
| Psychoanaleptics (N06) | 7.8 | 7.7 | NS |
| Other nervous system drugs (N07) | 7.9 | 5.1 | NS |

*P<0.05, **P<0.01, ***P<0.001.
ATC, Anatomical Therapeutic Chemical Classification System; NS, non-significant.

issue regarding sex differences in hospital readmission rates is the study window. Some studies found higher rates among men than among women below 3-month readmissions. More extended time windows (eg, 1 year) revealed no significant sex differences.[54 60] An analysis of our data set using a more extended readmission window might clarify this point and provide complementary knowledge about sex-associated hospital readmissions.

**Table 4** Baseline multilevel logistic regression model using 30-day readmission (0=no; 1=yes) as the dependent variable associated with independent sociodemographic, LOS and clinical variables (N=13 802)

| Variables | OR* | P value >z | 95% CI |
|---|---|---|---|
| Sex | 1.079 | 0.285 | 0.938 to 1.242 |
| Year-end age, in years | 0.999 | 0.878 | 0.990 to 1.009 |
| Hospital LOS, in days | 1.014 | 0.000 | 1.006 to 1.021 |
| Mobility cluster† | 1.218 | 0.015 | 1.039 to 1.427 |
| Dependency on the activities of daily living† | 0.794 | 0.248 | 0.537 to 1.174 |
| Mental health status† | 0.992 | 0.966 | 0.687 to 1.433 |
| ICD 1st diagnosis: circulatory problems‡ | 0.938 | 0.491 | 0.783 to 1.124 |
| ICD 1st diagnosis: infection‡ | 1.381 | 0.078 | 0.964 to 1.977 |
| ICD 1st diagnosis: respiratory problems‡ | 1.100 | 0.414 | 0.875 to 1.382 |
| ICD 1st diagnosis: trauma‡ | 0.847 | 0.265 | 0.633 to 1.134 |
| ICD 1st diagnosis: tumour‡ | 2.538 | 0.000 | 2.089 to 3.082 |
| Number of ICD | 1.419 | 0.000 | 1.282 to 1.572 |
| Number of CHOP | 0.978 | 0.304 | 0.938 to 1.020 |
| Number of drugs | 1.043 | 0.000 | 1.028 to 1.058 |
| Year: 2015–2018 | 0.933 | 0.022 | 0.880 to 0.990 |
| Intercept | – | 0.027 | – |

*Adjusted OR.
†0=good state; 1=impairment.
‡0=no; 1=yes.
CHOP, Swiss Classification of Surgical Interventions; LOS, length of stay.

Our results indicated that ageing was not a risk factor for increased 30-day hospital readmission, in line with some previous publications.[55 61] However, other research findings demonstrated that age was only positively associated with the likelihood of readmission up to 74 years old; above this, there no longer appeared to be any significant relationship between age and readmission.[62 63] These contrasting results may be explained by the studies' designs, country settings, age of research populations or the medical conditions included.[55 62 64]

Longer hospital stays were also associated with a higher risk of hospital readmission, in line with a cohort study by Sud *et al*[65] concluding that an extended hospital LOS was associated with increased rates of all types of readmission, except for hospitalisation after heart failure, where a short LOS was associated with increased rates of readmission for cardiovascular disease and heart failure.

Our results indicated a significant positive association between the number of a patient's medical conditions and the 30-day hospital readmission rate, confirming other recent retrospective hospital register studies.[66 67] More specifically, older patients with impaired mobility showed an increased risk of hospital readmission. This result was not surprising, bearing in mind that although these older patients were discharged home—and not to a nursing home—after their hospital stay, their health status might nevertheless require future readmission. Indeed, this corroborated publications about older patients discharged after orthopaedic treatment or who had been initially admitted for heart failure, myocardial infarction or pneumonia, but also presented with impaired mobility.[68 69]

Cognitive impairment was not associated with increased 30-day hospital readmission rates, in line with findings from the systematic review by Pickens *et al*,[70] which pointed out that dementia had a modest impact on readmission rates. It was no surprise that inpatients hospitalised for cancer faced a high risk of readmission, corroborating prior studies by Burhenn *et al*, Chang *et al* and Butcher.[71–73]

PP significantly increased the 30-day hospital readmission rate, but this result was based on the average number of drugs prescribed to the sample of readmitted patients versus those not readmitted. Although PP was confirmed as a strong determinant of 30-day hospital readmission in publications by Leendertse *et al*,[74 75] our results showed a progressive linear relationship between PP and readmission rate, and this should be interpreted with caution. Despite our results and other publications and research underlining the challenge of PP among multimorbid older patients, there is no overall consensus about the best way to deal with the broad general relationship between PP and hospital readmission.[76]

Our advanced statistical analysis demonstrated that some specific drugs and the concomitant use of specific drug combinations were significantly associated with 30-day readmission risk, although this was not unexpected and has been confirmed in previous publications.[37 77] Mostly in line with the research findings of Zhang *et al*, drugs including hormones, antineoplastics, immunosuppressors, neoplastic antibiotics and bacterial vaccines were substantial risk factors for hospital readmission.[7]

In summary, extended hospital LOS, functional impairments, medical conditions and drugs have been demonstrated to be determinants of 30-day hospital readmission, although not all of them have clinically or pharmacologically relevant interpretations or explanations. Further research involving large samples is needed, notably to explore the drug–drug interactions with the highest risk of hospital readmissions. Statistical predictions of potential drug–drug interactions provide important information for modelling drug combinations and identifying pairs of drugs whose combination creates an exaggerated response.[9] As the association between the number of drugs and the risk of hospital readmission was linear, more advanced inferential statistics would be needed to clarify a cut-off point for the mean number of drugs that would significantly increase the readmission rate. In addition, problems involving adherence to prescriptions, social support networks, and stronger or weaker

**Table 5** Multilevel logistic regression model results for the drugs prescribed to older patients at discharge home that had significant predictive values (OR) for 30-day hospital readmission (controlled for variables in the baseline model: table 4) (N=13 802)

| Variables | OR* | P value >z | 95% CI |
|---|---|---|---|
| First level, anatomical main group | | | |
| Blood and blood-forming organs drugs (B) | 1.089 | 0.041 | 1.003 to 1.181 |
| Systemic hormonal preparations, excluding sex hormones and insulins (H) | 1.207 | 0.007 | 1.052 to 1.385 |
| Respiratory system drugs (R) | 1.146 | 0.003 | 1.046 to 1.254 |
| Second level, therapeutic subgroup | | | |
| Drugs for functional gastrointestinal disorders (A03) | 1.424 | 0.001 | 1.166 to 1.739 |
| Antiemetics and antinauseants (A04) | 3.216 | 0.000 | 1.842 to 5.617 |
| Drugs for constipation (A06) | 1.195 | 0.018 | 1.031 to 1.386 |
| Drugs used in diabetes (A10) | 1.125 | 0.021 | 1.018 to 1.243 |
| Vitamins (A11) | 1.201 | 0.008 | 1.049 to 1.374 |
| Antihypertensives (C02) | 1.771 | 0.000 | 1.287 to 2.438 |
| Diuretics (C03) | 1.149 | 0.024 | 1.018 to 1.296 |
| Beta-blocking agents (C07) | 1.156 | 0.040 | 1.007 to 1.327 |
| Lipid-modifying agents (C10) | 0.841 | 0.015 | 0.732 to 0.967 |
| Psycholeptics (N05) | 1.130 | 0.009 | 1.031 to 1.238 |

*Adjusted OR.

primary healthcare structures can all influence hospital readmission rates.[39] According to some publications, nearly 70% of people aged over 65 make mistakes with their drugs.[78 79] Information about drug adherence, drug underuse and overuse, drug changes and deprescription by family physicians, as well as medication management at home would contribute to a more comprehensive understanding of disease-related and drug-related 30-day hospital readmissions.

Finally, it would be interesting to explore the risk of readmission according to different hospital wards. As psychiatric conditions are a frequent cause of rehospitalisation,[80] it would be relevant for future research to explore registries from adult psychiatry departments and investigate the hospital readmission risk faced by their inpatients.

**Strengths and limitations**
This study's main strength was its use of data recorded in a comprehensive register. We consider this retrospective study useful for clinical practice and future research because a whole series of sociodemographic and clinical parameters, medical conditions, and prescribed drugs

**Table 6** Drugs and drugs interactions from ATC classes A and B with a significant risk of 30-day hospital readmission (controlled for variables in the baseline model: table 4) (N=13 802)

| Variables | OR* | P value >z | 95% CI |
|---|---|---|---|
| First level, anatomical main group | | | |
| Blood and blood-forming organ drugs (B) | 1.089 | 0.040 | 1.004 to 1.182 |
| Systemic hormonal preparations, excluding sex hormones and insulins (H) | 1.210 | 0.007 | 1.054 to 1.390 |
| Respiratory system drugs (R) | 1.149 | 0.003 | 1.049 to 1.258 |
| Second level, therapeutic subgroup | | | |
| Antiemetics and antinauseants (A04) | 3.222 | 0.000 | 1.844 to 5.630 |
| Drugs for functional gastrointestinal disorders (A03) | 1.428 | 0.000 | 1.169 to 1744 |
| Beta-blocking agents (C07) and drugs for acid-related disorders (A02) | 1.367 | 0.022 | 1.046 to 1.788 |
| Drugs for constipation (A06) | 1.199 | 0.017 | 1.033 to 1.392 |
| Agents acting on the renin–angiotensin system (C09) | 0.892 | 0.049 | 0.796 to 0.999 |
| Lipid-modifying agents (C10) | 0.838 | 0.013 | 0.729 to 0.964 |

*Adjusted OR.
ATC, Anatomical Therapeutic Chemical Classification System.

were used to predict the probability of hospital readmission. Using both bivariate and multivariate analyses enabled an evaluation of the data's longitudinal nature.

Our study had several limitations, nevertheless. The design did not allow us to identify hospitalisations and readmissions lost to follow-up and to adjust our data for death outside the hospital. We were also unable to identify unnecessary hospitalisations or any bias towards hospitalisation rather than another healthcare solution for older inpatients. Our data set could not inform us about whether older inpatients had been first admitted to another hospital or were subsequently readmitted elsewhere during the study period. Because the reasons for hospital admission are not chosen from a list but are entered into the register as free descriptive text, these factors were not part of our data set, and the study was unable to explore the reasons for an admission's impact on 30-day rehospitalisation. Another limitation was the study's lack of formal screening methods to explain ADEs in detail, and it was impossible to distinguish between elective and urgent hospitalisations. Although the study considered statistical associations between drugs and rehospitalisations, it did not use clinically diagnosed drug–drug interactions. Finally, we were unable to consider any potential causality between PP and hospital readmission.

## CONCLUSIONS

Hospital LOS, medical conditions, functional impairments and prescribed drugs were all critical factors in predicting hospital readmissions, thus affirming our hypotheses. Readmission patterns are complex and poorly understood because older patients often present with multiple chronic conditions, functional impairments and complex drug prescriptions. Hospital readmission is an underinvestigated topic deserving of additional, well-conducted, predictive research exploiting accurate longitudinal data from large samples.

**Author affiliations**
[1]Institute of Biomedical Sciences Abel Salazar, University of Porto, Porto, Portugal
[2]School of Health Sciences, HES-SO University of Applied Sciences and Arts Western Switzerland, Sion, Switzerland
[3]Service of Old Age Psychiatry, Lausanne University Hospital, Lausanne, Switzerland
[4]Faculty of Psychology and Educational Sciences, University of Geneva, Geneva, Switzerland
[5]Pharmacy Benu Tavil-Chatton, Morges, Switzerland
[6]Institute for Primary Health Care, University of Bern, Bern, Switzerland
[7]Porto Higher School of Nursing, Porto, Portugal
[8]FORS, Swiss Centre of Expertise in the Social Sciences, University of Lausanne, Lausanne, Switzerland

**Acknowledgements** The authors thank the partner hospital, including the hospital's data warehouse, for its valuable contributions.

**Contributors** BW, FP and HV had the original idea. BW, ZT, SDG, MMM and HV provided conceptual and methodological expertise to the study design and BW, FP, ZT, SDG, CM-M, AV-G and HV to data analysis and interpretation. BW, FP and HV were major contributors to writing the manuscript. All authors read, edited and approved the final manuscript.

**Funding** This study was supported by the Swiss National Science Foundation via grant number 407440_183434/1. This research was developed, in part, using grants from the Swiss National Science Foundation and the School of Health Sciences of the University of Applied Sciences and Arts Western Switzerland (HES-SO) Valais/Wallis.

**Disclaimer** The funders had no role in the design and conduct of the study, collection, management, analysis and interpretation of data, preparation, review or approval of the manuscript, or the decision to submit the manuscript for publication.

**Competing interests** None declared.

**Patient consent for publication** Not required.

**Ethics approval** Ethical approval was obtained from the Human Research Ethics Committee of the Canton of Vaud (CER-VD, 2018-02196), thus permitting our partner hospital's data warehouse to provide the appropriate data set. Given the retrospective data source, obtaining consent from the patients concerned was impossible or posed disproportionate difficulties. The present study respects the legal requirements for research projects involving data reuse without consent, as set out in Art 34 from the Swiss Human Research Act (HTA).

**Provenance and peer review** Not commissioned; externally peer reviewed.

**Data availability statement** Data are available upon reasonable request.

**ORCID iDs**
Filipa Pereira http://orcid.org/0000-0001-9207-4856
Henk Verloo http://orcid.org/0000-0002-5375-3255

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
