## [Reviewer comments · BMJ Open]

ARTICLE DETAILS

TITLE (PROVISIONAL)	Risks of 30-day hospital readmission associated with medical conditions and drug regimens of polymedicated, older inpatients discharged home: a registry-based cohort study
AUTHORS	Pereira, Filipa; Verloo, Henk; Zhivko, Taushanov; Di Giovanni, Saviana; Meyer-Masseti, Carla; Von-Gunten, Armin; Martins, Maria Manuela; Wernli, Boris

VERSION 1 – REVIEW

REVIEWER	Fabbietti, Paolo INRCA
REVIEW RETURNED	14-May-2021

GENERAL COMMENTS	for me it's good.
-------------------

REVIEWER	Sharma, Manvi University of Mississippi
REVIEW RETURNED	18-May-2021

GENERAL COMMENTS	Overall Comments: The manuscript presented attempts to address an important research question. However, there are several concerns regarding the write-up. Overall, the objectives, methods and results section do not appear to be consistent with each other. One overarching concern is the heterogeneity of the population examined – the patients included in the study are quite heterogenous in terms of hospital admissions to make any meaningful conclusions. The authors do not make a distinction between the reason for primary hospital admission which can have huge impact on the re-admission. The use of the term “risk” through the manuscript needs to be evaluated for correct interpretation. There are several methodological concerns. These are outlined below. Specific comments: Page 4, line 2-4: The use of term “risk” in the objective statement is somewhat inaccurate as the authors did not investigate the association between re-admission risk and the patient characteristics but determined the risk of re-admission associated with several characteristics. This may be a linguistic difference but the way the statement is written, it is not clear. Page 4, line 5: Design: If the study design is a retrospective cohort, it would add clarity to specify here. Page 4, line 12-18: the statement “higher risks of hospital
--

	readmission were associated with longer hospital length of stay” is confusing- how was risk of hospital re-admission measured? Based on the methods reported, the ORs report the risk of hospital re-admission among patients with variable length of stay. The interpretation of these results would be – an increased risk of hospital re-admission was found for patients with longer hospital length of stay. Please clarify the interpretations of the ORs reported. Same comment for the other ORs reported in the results section. This may again be a linguistic difference but the way the statement is written, it is not clear. Page 4, line 12: please include that the results reported i.e ORs are multivariable / adjusted for the variables included in the final model. Page 4, line 19: same comment as above regarding the term “risk” associated with various characteristics. Page 7, line 118-121: the inclusion and exclusion criteria needs to be clearly described in this section. Did the patients have to be polymedicated i.e. patient discharged with less than 5 medications were not included in the study? Clarify identification of multi-morbid condition – if any two ICD codes were used to categorize patients as multi-morbid – how valid was this measure? Could the two ICD codes be for same/similar condition and be double counted? Does this classification distinguish acute vs chronic conditions? “Patients had to be admitted and readmitted” – what does this mean? Does it mean that all the patients had to have 1 hospital admission or more than 1 hospital admission? What was the exclusion criteria? Were there more than 2 (multiple) hospitalizations for any patients? how were these treated? Was the primary diagnosis for baseline hospitalization taken into consideration? If ye, please describe how? If no, please describe why not? Page 7, line 129-131: although authors cite the reference #47 for details of data collection, it would be helpful to include a small summary here for clear understanding of the chart review process and extraction of variables from different sources within the electronic medical records. Need more clear description of what data is available in the hospital discharge records – eg how many chronic conditions are listed etc? What is the implication of missing data for psychiatry? What does exhaustive register mean? Please describe. A timeline of collection of different measures may be helpful to clearly understand the process of data collection. Page 8, line 156: this seems to be a result and may be better placed in the results section and not methods section. Page 8, line 161-164: why were three of the six clusters selected? Page 8, section “Health variables”: this section seems to include more of the results rather than methods to measure the variables. The results would be better reported in results section unless these results were utilized to make decisions for inclusion of the patients and/or variables for the study. If so, please clarify and report potential biases introduced due to such selection criteria. Page 8, line 169-172: if the primary outcome evaluated for the study
--	--

	is 30-day readmission, it is not clear what is addressed by using the year of hospitalization. Was every patient followed for 30-days post discharge? Were some patients followed for longer? The sentence - “unplanned readmissions during the entire period covered” raises several more questions regarding the methods for this study. Were the 30-day readmissions assessed planned or unplanned? Was the period covered to assess the outcome different for different patients? Page 8, line 191: What do authors mean by “multilevel binary logistic regression models”? Were hierarchical regression models utilized to account for nesting? Were the patients considered nested within LOS categories? Page 9, line 200-202: How is the study cohort considered whole population when there were several inclusion criteria including the number of medications used etc. Page 9, line 204-206: this seems to be repeated. (same as page 7, lines 135-137) Page 9, line 211: Please justify the use of term “prevalence” for reporting the result. Please review the use of this term in the remainder of the results as well. Page 9, line 211-213: What statistical tests were used for the bivariate associations? Page 10, line 235: The term “multivariable” may be more appropriate since “multivariate” is often used to denote multiple dependent variables. What was the modeling strategy utilized for the multivariable logistic regression? What was the basis for inclusion and exclusion of the variables into the baseline model? Why were prescribed drugs excluded from this model? The use of term “predict” seems problematic as the goal of the study has been to determine association between patient characteristics and hospital re-admission. Please review the use of term “predict/prediction” throughout the manuscript and be consistent. Page 10, line 247: “several pathologies” is a new terminology introduced in the results section. Please either include this in methods or use the terms described in the methods. Page 10, line 270: “Drug interactions” have not been mentioned in the methods section at all. It is not acceptable to introduce results that are not planned or reported as per the methods section. Please discuss results of table 2 and table 4. As per table 2: most factors are NS in bivariate but several are significant in multivariate analysis (table 4). Please discuss why? Was information on wards/units hospitalized to assessed and included? Like jencks study? Table 1: What is the inference from this table? This table does not add much value as the first table in the study. This could be a supplementary table or an appendix. Ideally table 1 describes the baseline population characteristics. Page 13, line 372: The lack of information on death, and potential limitation of loss of follow-up is a critical limitation and thus affects what can be inferred from the results of the study. How many were
--	--

	lost to follow-up – the comparison group for determining the risk of readmission is questionable.
--	---

VERSION 1 – AUTHOR RESPONSE

Reviewer: 1
Dr. Paolo Fabbietti, INRCA

Reviewers' 1 comments	Response by authors	Location in text
For me it's good [NO ADDITIONAL COMMENTS]	We thank the reviewer for his support.	

Reviewer: 2
Dr. Manvi Sharma, University of Mississippi Comments to the Author: [PLEASE SEE ATTACHED PDF FILE FOR REVIEWER COMMENTS]
Note: Lekha Negi, PharmD, BCGP collaborated on reviewing this manuscript.

Reviewers' 2 overall comments	Response by authors	Location in text
Overall comments: The manuscript presented attempts to address an important research question. However, there are several concerns regarding the write-up.	We thank the reviewer for their support. We have made every effort to respond, correct and adapt to the comments provided.	Whole manuscript.
Overall, the objectives, methods and results section do not appear to be consistent with each other.	These sections have been revised and updated.	Whole manuscript.
One overarching concern is the heterogeneity of the population examined – the patients included in the study are quite heterogenous in terms of hospital admissions to make any meaningful conclusions.	Indeed, hospitals admit older adults with many different profiles, including patients requiring complex healthcare management. The heterogeneity of the population examined is explained by our study's objectives: to determine the risks of 30-day hospital readmissions based on their (multitudinous) medical conditions and the (huge variety of) drug regimens that polymedicated, older inpatients discharged home are prescribed. Thus, our study was based on routinely collected "real-life" data from the Valais Hospital. Our paper provides evidence that can encourage new research perspectives on the hospital readmissions of polymedicated home-dwelling older inpatients. Our multidisciplinary international research team believes that this is a fundamental step towards future research that may be able to drill down into the detail.	

The authors do not make a distinction between the reason for primary hospital admission which can have huge impact on the re-admission.	Regrettably, the reasons for hospital admission are entered into the register as free text and were thus unavailable in the dataset. Even though it is usual to have access to these data in a prospective study, it is not the case in a hospital register based on routine data. We have included this comment in the Strengths and Limitations section: "Because the reasons for hospital admission are not chosen from a list but are entered into the register as free descriptive text, these factors were not part of our dataset, and the study was unable to explore the reasons for an admission's impact on 30-day rehospitalisation."	Strengths and Limitations, lines 443-446
The use of the term "risk" through the manuscript needs to be evaluated for correct interpretation. There are several methodological concerns. These are outlined below.	We thank the reviewer for his suggestion. We have made every effort to correct the use of the term "risk".	Whole manuscript
Reviewers' 2 specific comments	Response by authors	Location in text
Page 4, line 2-4: The use of term "risk" in the objective statement is somewhat inaccurate as the authors did not investigate the association between re-admission risk and the patient characteristics but determined the risk of re-admission associated with several characteristics. This may be a linguistic difference but the way the statement is written, it is not clear.	Indeed, it is more accurate to state the objective as suggested. Below are the adjustments we have made. "Objectives: The present study analysed four years of a hospital register (2015–2018) to determine the risks of 30-day hospital readmission associated with the medical conditions and drug regimens of polymedicated, older inpatients discharged home." "This longitudinal study (2015–2018) used data on a population cohort taken from a hospital registry composed of 140 variables. These were used to investigate the associations between the risks of 30-day hospital readmission and the medical conditions and drug regimens of polymedicated older inpatients discharged home."	Abstract, lines 2-4 Study Design, lines 119-123
Page 4, line 5: Design: If the study design is a retrospective cohort, it would add clarity to specify here.	We have clarified the design as "registry-based cohort study" in different sections. We thank the reviewer for this important comment.	Title Abstract, line 5 Study design, lines 119-125
Page 4, line 12-18: the statement "higher risks of hospital readmission were associated with longer hospital length of stay" is confusing- how was risk of hospital re-admission measured? Based on the methods reported, the ORs report the risk of hospital re-admission among patients with variable length of stay. The	Thank you for alerting us to these relevant subtleties. We have replaced "Higher risks of hospital readmission were associated with longer hospital length of stay" with "Adjusted multivariate analyses revealed increased risks of hospital readmission for patients with longer hospital lengths of stay (...), impaired mobility (...), multimorbidity (...), tumoural disease (...),	Abstract, results, lines 12-25

interpretation of these results would be – an increased risk of hospital re-admission was found for patients with longer hospital length of stay. Please clarify the interpretations of the ORs reported. Same comment for the other ORs reported in the results section. This may again be a linguistic difference but the way the statement is written, it is not clear.	polypharmacy (...), and certain specific drugs (...).”	
Page 4, line 12: please include that the results reported i.e ORs are multivariable / adjusted for the variables included in the final model.	We have completed our results as follows: “Adjusted multivariate analyses revealed increased risks of hospital readmission for patients with longer hospital lengths of stay (...), impaired mobility (...), multimorbidity (...), tumoural disease (...), polypharmacy (...), and certain specific drugs (...).”	Abstract, line 12
Page 4, line 19: same comment as above regarding the term “risk” associated with various characteristics.	As mentioned above, we have integrated this comment as follows: Adjusted multivariate analyses revealed increased risks of hospital readmission for patients with longer hospital lengths of stay (...), impaired mobility (...), multimorbidity (...), tumoural disease (...), polypharmacy (...), and certain specific drugs (...).”	Abstract, lines 12-25
Page 7, line 118-121: the inclusion and exclusion criteria needs to be clearly described in this section. Did the patients have to be polymedicated i.e. patient discharged with less than 5 medications were not included in the study?	We have completed this section as follows: “Our custom, four-year, registry-based dataset included polymedicated inpatients (five or more drugs prescribed at hospital discharge), aged 65 years old or more, living in their own homes and hospitalised at least once at the Valais Hospital (a public general hospital in the French-speaking part of Switzerland).”	Population and Data Collection, line 130-135
Clarify identification of multi-morbid condition – if any two ICD codes were used to categorize patients as multi-morbid – how valid was this measure? Could the two ICD codes be for same/similar condition and be double counted? Does this classification distinguish acute vs chronic conditions?	Multimorbidity was not an inclusion criterion (even though, from a clinical point of view, it is difficult to be polymedicated without multimorbidity).	
“Patients had to be admitted and readmitted” – what does this mean? Does it mean that all the patients had to have 1 hospital admission or more than 1 hospital admission?	Indeed “admitted and readmitted” is confusing. To avoid ambiguity, we have replaced by “hospitalised at least once”.	Population and Data Collection, line 131-132
What was the exclusion criteria?	In response to this comment, we have added the exclusion criteria in the manuscript: “Older inpatients hospitalised once only or who died during hospitalisation were excluded, as were those hospitalised for fewer than 24 hours (the criterion to count as “hospitalised” in Switzerland).”	Population and Data Collection, lines 134-137
Were there more than 2 (multiple) hospitalizations for any patients? how were these treated?	Yes, our analyses looked at those who were hospitalised at least twice between 2015 and 2018 (8,878 different older inpatients discharged home and readmitted at least	Whole manuscript

	once). This has been better described throughout the manuscript and particularly in the inclusion and exclusion criteria.	
Was the primary diagnosis for baseline hospitalization taken into consideration? If ye, please describe how? If no, please describe why not?	The primary ICD-10 of the ongoing hospitalisation was considered. However, the relationship with the main ICD-10 of the rehospitalisation was not addressed because our dataset included more than 2,000 different primary diagnoses (as already mentioned, our subjects had very heterogeneous profiles). Using every different primary diagnosis for baseline hospitalisation and calculating their overall readmission rates was not considered in this study and seemed unfeasible with regards to our research question. However, the reviewer's comment could form part of a next step in our analysis of our large dataset, based on single primary diagnoses or diagnostic classes (e.g. ICD-10 diagnostic groups F.00 to F.99).	
Page 7, line 129-131: although authors cite the reference #47 for details of data collection, it would be helpful to include a small summary here for clear understanding of the chart review process and extraction of variables from different sources within the electronic medical records. Need more clear description of what data is available in the hospital discharge records – eg how many chronic conditions are listed etc?	We have updated the description of the available data as suggested. “Valais Hospital’s register contains a comprehensive electronic health record composed of 140 variables routinely collected during hospital stays.” (lines 139-141) “The sociodemographic data set—almost exclusively composed of ordinal variables— included two categorical variables (sex and place of discharge from hospital) and three continuous variables (age and admission and discharge dates).” (lines 174-176) “The health dataset was composed of 23 categorical variables: 21 measured as ordinal variables (mobility, changing position, falls in the last year, etc.) and two measured as nominal variables (altered gait and chronic pain). A cleaner, better-structured dataset - composed of hierarchical clusters - was obtained in a previous study, combining empirical and best-practice statistical approaches (47).” (lines 183-187) “The hospital dataset showed that discharged patients had been prescribed 2,370 different drugs. Drug prescriptions were considered continuous, classified according to the WHO’s ATC Classification System (49) and then included in the predictive model as independent variables. ” (lines 203-205)	Methods section, lines 139-196
What is the implication of missing data for psychiatry?	Indeed, no electronic patient records were available from the adult psychiatry department for that period. We have examined this point in the discussion section with the	Discussion, lines 429-431

	following sentence: “As psychiatric conditions are a frequent cause of rehospitalisation (78), it would be relevant for future research to explore registries from adult psychiatry departments and investigate the hospital readmission risks faced by their inpatients.”	
What does exhaustive register mean? Please describe.	To avoid misunderstanding, we have replaced the term “exhaustive” by “comprehensive”.	Whole manuscript.
A timeline of collection of different measures may be helpful to clearly understand the process of data collection.	As described in the Material and Methods section: “Valais Hospital’s register contains a comprehensive electronic health record composed of 140 variables routinely collected during hospital stays.” All data were routinely collected throughout hospitalisations. After approval by the ethics committee in January 2019, a request to extract the register dataset was submitted to the Valais hospital. The register dataset was received on 15 September 2019. Different data managers cleaned-up, transformed and customised the register dataset from October 2019 to January 2020 (1) and ongoing analysis started on February 2020. The present publication is the first step in our analysis of the large dataset. From 2021 to 2024 we will be exploring disease-based and drug-based analyses using different multivariate linear and logistic models.	Material and Methods, lines 139-141
Page 8, line 156: this seems to be a result and may be better placed in the results section and not methods section.	We have moved these items to a new section of the results entitled “descriptive results”.	Methods and Results, lines 253-265
Page 8, line 161-164: why were three of the six clusters selected?	We have updated this section as follows: “These three clusters were selected because of their significant contributions to hospital readmissions (49-51).”	Material and Methods, lines 188-189
Page 8, section “Health variables”: this section seems to include more of the results rather than methods to measure the variables. The results would be better reported in results section unless these results were utilized to make decisions for inclusion of the patients and/or variables for the study. If so, please clarify and report potential biases introduced due to such selection criteria.	We have moved these items to a new section of the results entitled “descriptive results”.	Methods and Results, lines 253-265
Page 8, line 169-172: if the primary outcome evaluated for the study is 30-day readmission, it is not clear what is addressed by using the year	As addressed on lines 196-198, “the year of hospitalisation was introduced as a control variable, based on the fact that earlier admission to hospital during this period led to a	Lines 196-198

of hospitalization. Was every patient followed for 30-days post discharge? Were some patients followed for longer? The sentence - “unplanned readmissions during the entire period covered” raises several more questions regarding the methods for this study. Were the 30-day readmissions assessed planned or unplanned? Was the period covered to assess the outcome different for different patients?	higher probability of unplanned readmissions during the entire period covered.” We highlight that our study was based on extracting data from a hospital patient register over a 4-year period. Based on the available data, we could not discriminate between planned or unplanned hospitalisations due to the fact that electronic data from the emergency department is not connected to the electronic hospital register. Some studies have mentioned that readmission rates should be based on an 18-month period before the end date of the register. However, for datasets shorter than 5 years, this is not at all recommended. To counter the longitudinal barrier of our 4-year dataset, we planned to conduct a survival analysis to clarify the probability of readmission or unplanned institutionalisation after each second hospitalisation (=readmission) in the dataset.	
Page 8, line 191: What do authors mean by “multilevel binary logistic regression models”?	We agree that our description of the logistic regressions in the analysis section may have lacked a little clarity. We have made the following revision: “Next, we calculated a series of multilevel logistic regression models for binary outcomes explaining the readmissions, within 30 days, of patients discharged home (0 = no, 1 = yes). These hierarchical models included two levels: the first level concerned hospital stays themselves, nested in the second level, that of individuals. Firstly, we computed a baseline multilevel binary logistic regression model to estimate how sets of predictors influenced the probability of 30-day hospital readmission, which included individuals’ characteristics, health conditions and hospital LOS. Secondly, we completed this baseline model with the drugs prescribed to older inpatients on their discharge home and at their 30-day hospital readmission. Finally, to that baseline model completed with prescribed drugs, we added the known drug–drug interactions between different ATC drug classes, based on a literature review and expert opinions.”	Data analysis strategy, lines 222-233
Were hierarchical regression models utilized to account for nesting?	Yes, through our multilevel models, where the information is nested.	
Were the patients considered nested within LOS categories?	No, hospital LOS was considered for all hospitalisations (N = 20,422) and not by categories (such as age group or another classification).	
Page 9, line 200-202: How is the study cohort considered whole population when there were several inclusion criteria including the number of medications used etc.	To address this issue, we have completed the text as follows: “Since the data were based on the whole population—not a sample—of polymedicated older inpatients discharged	Data analysis strategy, lines 240-243

	home from the Valais Hospital, the ORs' confidence intervals and statistical tests were used to indicate the robustness of relationships (they usually only make sense for statistical inference)."	
Page 9, line 204-206: this seems to be repeated. (same as page 7, lines 135-137)	Indeed, it was a duplicate. We thank the reviewer for noticing it.	Lines 152-154
Page 9, line 211: Please justify the use of term "prevalence" for reporting the result. Please review the use of this term in the remainder of the results as well.	We agree with the reviewer that the term "prevalence" was not well applied. Therefore, we have replaced it by "rate" and "risk" throughout the manuscript.	Whole manuscript
Page 9, line 211-213: What statistical tests were used for the bivariate associations?	We have completed the manuscript as follows: "Bivariate associations with chi-square tests showed significant differences between..."	Results, line 270
Page 10, line 235: The term "multivariable" may be more appropriate since "multivariate" is often used to denote multiple dependent variables. What was the modeling strategy utilized for the multivariable logistic regression? What was the basis for inclusion and exclusion of the variables into the baseline model? Why were prescribed drugs excluded from this model?	Based on Hidalgo and Goodman (2), the term "multivariate" analysis seems to be an appropriate adjective for this study.	Whole manuscript
The use of term "predict" seems problematic as the goal of the study has been to determine association between patient characteristics and hospital re-admission. Please review the use of term "predict/prediction" throughout the manuscript and be consistent.	A risk prediction model was used to measure associations. As predictive models were computed to achieve the study's objective, we believe that it is important to maintain the term "predict"(3).	Whole manuscript
Page 10, line 247: "several pathologies" is a new terminology introduced in the results section. Please either include this in methods or use the terms described in the methods.	We have replaced "several pathologies" by "multimorbidity" to remain consistent with the other sections.	Results, line 306
Page 10, line 270: "Drug interactions" have not been mentioned in the methods section at all. It is not acceptable to introduce results that are not planned or reported as per the methods section.	Indeed, these interactions should have been described in the methods section. To address this issue, we have moved the paragraph down from the results to the methods: "For statistical purposes, drug–drug interactions between different ATC drug classes (49) were operationalised as dichotomised variables (0 = no simultaneous use of drugs from both classes, 1 = simultaneous use of drugs from both classes) and added to the previous model. Drug class interactions were selected based on a literature review, significant ORs and expert opinions (50)."	Methods, lines 209-212
Please discuss results of table 2 and	We have added the following paragraph:	Results,

table 4. As per table 2: most factors are NS in bivariate but several are significant in multivariate analysis (table 4). Please discuss why?	“Some variables that were non-significant in bivariate analyses became significant in multivariate analyses. This was because the results of multivariate analyses were controlled by all the other parameters and interpretations were made with “other things being equal”. Also, the composition of subgroups could be very different in some bivariate analyses.” We take up and discuss the most relevant results in the concluding section.	lines 311-314
Was information on wards/units hospitalized to assessed and included? Like jencks study?	No, we did not distinguish between the different hospital wards, but this may be relevant for future research. We have introduced this element into the discussion section: “Finally, it would be interesting to explore the risks of readmission according to different hospital wards.”	Discussion, line 428
Table 1: What is the inference from this table? This table does not add much value as the first table in the study. This could be a supplementary table or an appendix. Ideally table 1 describes the baseline population characteristics.	We have moved the original Table 1 to Supplementary File 1 and added a new Table 1 with a description of the characteristics of the patients readmitted to the hospital.	Results, line 265
Page 13, line 372: The lack of information on death, and potential limitation of loss of follow-up is a critical limitation and thus affects what can be inferred from the results of the study. How many were lost to follow-up – the comparison group for determining the risk of readmission is questionable	Indeed, this is a critical limitation, as mentioned in the “Strengths and Limitations” section: “The design did not allow us to identify hospitalisations and readmissions lost-to-follow-up and to adjust our data for death outside the hospital.” However, we would point out that we did control for health status, age and type of disease, parameters that naturally have an impact on mortality and are therefore also included in the model. The following points offer some explanations as to why our register is limited to document deaths and loss of follow-up:  1. Death is one of the polytomous items making up the variable of “destination” after hospitalisation. The cause of death is not registered in the hospital data base. 2. Sudden deaths in Valais Hospital, mostly caused by fatal cardiovascular events or trauma, are documented in the register, but palliative care is not and end of life is mostly managed at home or in long-term or specialised care facilities. 3. Sudden death occurring in the emergency department is not included in the hospital register. 4. Finally, the hospital register gave us no information on death rates. 5. Older adult hospitalisations in 	Strengths and Limitations, lines 438-439

	public or private hospitals in other cantons were not documented in our hospital register. Future research should explore other data resources (public health statistics and cantonal and national death registers) to accurately document death rates, causes of deaths and losses to follow-up in other public and private hospitals in other cantons.	
--	--	--

1. Taushanov Z, Verloo H, Wernli B, Giovanni SD, Gunten Av, Pereira F. Transforming a Patient Registry Into a Customized Data Set for the Advanced Statistical Analysis of Health Risk Factors and for Medication-Related Hospitalization Research: Retrospective Hospital Patient Registry Study. *JMIR Medical Informatics*. 2021;9(5).
2. Hidalgo B, Goodman M. Multivariate or multivariable regression? *American journal of public health*. 2013;103(1):39-40.
3. Palmer PB, O'Connell DG. Regression analysis for prediction: understanding the process. *Cardiopulmonary physical therapy journal*. 2009;20(3):23.